# Pancreatic Cancer Organoids in the Field of Precision Medicine: A Review of Literature and Experience on Drug Sensitivity Testing with Multiple Readouts and Synergy Scoring

**DOI:** 10.3390/cancers14030525

**Published:** 2022-01-21

**Authors:** Lotta Mäkinen, Markus Vähä-Koskela, Matilda Juusola, Harri Mustonen, Krister Wennerberg, Jaana Hagström, Pauli Puolakkainen, Hanna Seppänen

**Affiliations:** 1Department of Surgery, Faculty of Medicine, University of Helsinki and Helsinki University Hospital, 00290 Helsinki, Finland; lotta.makinen@helsinki.fi (L.M.); matilda.juusola@helsinki.fi (M.J.); harri.mustonen@helsinki.fi (H.M.); pauli.puolakkainen@hus.fi (P.P.); 2Translational Cancer Medicine Research Program, Faculty of Medicine, University of Helsinki, 00290 Helsinki, Finland; 3Institute for Molecular Medicine Finland FIMM, Helsinki Institute for Life Science HiLIFE, University of Helsinki, 00290 Helsinki, Finland; markus.vaha-koskela@helsinki.fi (M.V.-K.); krister.wennerberg@bric.ku.dk (K.W.); 4Department of Pathology, Faculty of Medicine, University of Helsinki and Helsinki University Hospital, 00290 Helsinki, Finland; jaana.hagstrom@hus.fi; 5Department of Oral Pathology and Radiology, Faculty of Medicine, University of Turku, 20520 Turku, Finland

**Keywords:** pancreatic ductal adenocarcinoma, organoid, cancer precision medicine, drug combinations

## Abstract

**Simple Summary:**

New treatments are urgently needed for pancreatic ductal adenocarcinoma because it is one of the most aggressive and lethal cancers, detected too late and resistant to conventional chemotherapy. Tumors in most patients feature a similar set of core mutations but so far it has not been possible to design a one-fits-all treatment strategy. Instead, efforts are underway to personalize the therapies. To find the treatments that might work the best for each patient, entirely new experimental platforms based on living miniature tumors, organoids, have been developed. We review here the latest international findings in designing personalized treatments pancreatic cancer patients using organoids as testing beds. Our own work adds important clues about how such testing could, and perhaps should, be conducted.

**Abstract:**

Pancreatic ductal adenocarcinoma (PDAC) is a silent killer, often diagnosed late. However, it is also dishearteningly resistant to nearly all forms of treatment. New therapies are urgently needed, and with the advent of organoid culture for pancreatic cancer, an increasing number of innovative approaches are being tested. Organoids can be derived within a short enough time window to allow testing of several anticancer agents, which opens up the possibility for functional precision medicine for pancreatic cancer. At the same time, organoid model systems are being refined to better mimic the cancer, for example, by incorporation of components of the tumor microenvironment. We review some of the latest developments in pancreatic cancer organoid research and in novel treatment design. We also summarize our own current experiences with pancreatic cancer organoid drug sensitivity and resistance testing (DSRT) in 14 organoids from 11 PDAC patients. Our data show that it may be necessary to include a cell death read-out in ex vivo DSRT assays, as metabolic viability quantitation does not capture actual organoid killing. We also successfully adapted the organoid platform for drug combination synergy discovery. Lastly, live organoid culture 3D confocal microscopy can help identify individual surviving tumor cells escaping cell death even during harsh combination treatments. Taken together, the organoid technology allows the development of novel precision medicine approaches for PDAC, which paves the way for clinical trials and much needed new treatment options for pancreatic cancer patients.

## 1. Introduction

Pancreatic ductal adenocarcinoma (PDAC) is an aggressive malignancy with an overall 5-year survival rate of 9% [1]. At the national level in Finland, median survival in patients undergoing radical pancreatectomy increased from 20 months during the years 2000–2008 to 28 months in 2009–2016 [2]. While radical surgery serves as the only potentially curative treatment, the resulting 5-year overall survival only increases to 20–30%. Most patients (80–90%) are diagnosed with locally advanced or metastatic disease, with only 10–20% eligible for surgical resection. Patients with borderline disease are treated with neoadjuvant cytotoxic chemotherapy before surgical resection, with most receiving adjuvant chemotherapy [3]. With locally advanced or metastatic disease, the combination therapies of gemcitabine/nab-paclitaxel or FOLFIRINOX (5-fluorouracil, leucovorin, irinotecan and oxaliplatin) are used [4,5]. Neither of these chemotherapy combinations have emerged as superior to the other, but they each carry distinct adverse effect profiles, whereby FOLFIRINOX is considered more toxic [6,7].

PDAC is a rather heterogeneous disease with multiple oncogenic mutations. Over 90% of PDAC tumors have a mutation in oncogene *KRAS*. Other commonly mutated genes with study-dependent prevalence rates include tumor-suppressor genes CDKN2A (49–98%)*,* TP53 (20–76%) and SMAD4 (19–50%), along with many other gene alterations detected at a lower prevalence [8,9]. A single tumor may harbor numerous mutations, although variation between different tumors remains high. Target therapies have thus been difficult to develop. Of 1856 patients with PDAC enrolled in the Know Your Tumor personalized clinical trial, 282 patients harbored actionable mutations [10]. Of 189 patients with evaluable outcomes, 46 had an improvement of overall survival when given molecularly matched treatments compared to 143 patients receiving unmatched therapies [10]. Furthermore, only 4–7% of patients with BRCA1/2-mutated tumors have a marked response to treatment with platinum-based chemotherapy [8,11]. These findings argue for continued development of personalized treatment approaches that do not rely solely on genetic information.

More recent developments in PDAC tumor classification include the establishment of transcriptional typing, which has emerged as the preferred method for benchmarking personalized treatment efforts [12,13]. Seminal studies by Collisson [14], Moffitt [15] and Bailey [16] each attempted to categorize cases by their relative positioning along a flexible epithelial–mesenchymal axis, progressively accounting for the stromal components (Bailey enriched epithelial cells to over 40%). Two partially overlapping primary subtypes have emerged from these studies: (1) the classic/stable/progenitor subtype, and (2) the quasi-mesenchymal/basal/squamous subtype. Among these, “basal-like” tumors are less differentiated and are associated with worse prognoses than “classic” tumors [15,16]. A potential bias in the transcriptional signatures may result from the smaller number of metastatic tumors sampled compared with primary tumors.

Due to the tumor heterogeneity and a lack of target therapies, precision medicine has received much interest. In 1975, Wright and Walker [17], for instance, described one of the first experimental models to evaluate the differential potency of anticancer agents, selecting agents for different cancer types and/or patients—that is, precision medicine. In recent years, a new three-dimensional primary PDAC-culturing model termed patient-derived organoid (PDO) was developed [18]. PDOs serve as platforms for drug sensitivity and resistance testing (DSRT), which aids in the discovery of novel effective treatments for PDAC patients and can provide new potential diagnostic biomarkers for the disease.

In this article, we first review the current PDAC model systems, summarized in Figure 1, and focus on PDO models and their applications. In Section 5, we then present the technical development of and initial results from our own pancreatic cancer organoid-based DSRT.

## 2. Traditional PDAC Models

### 2.1. PDAC Cell Lines 

Pancreatic cancer cell lines represent a traditional means of studying cancer. The first PDAC cell line was established in 1963 [19]. Cell lines are easy to grow but carry many disadvantages and limitations in the study of PDAC. First, growing cells in a monolayer on plastic does not mimic the physical conditions in tumors, where cells are under heavy pressure and mechanical strain. Second, the cell lines lack stromal cells, thereby not reflecting the physiological conditions. Third, others have noted that during cell line establishment, more aggressive clones are selected, and cell lines acquire an increasing number of mutations with passage [20,21]. Furthermore, while the most commonly used commercial PDAC cell lines differ in phenotype and genotype [21], these and nearly all other PDAC cell lines studied represent the basal-like class, indicating that they represent only a specific disease subtype [15,16]. Accordingly, cell lines become poor representations of the original tumor specimen. This problem is compounded by the fact that normal pancreatic ductal cells are difficult to culture, further hampering comparison between healthy and malignant tissue samples.

PDAC cell lines can be cultured in 3D by preventing the cell from attaching to the bottom of the plate. The resulting 3D spheroids may mimic the physiological conditions better than a 2D model, allowing for model cell-to-cell interactions. Spheroids are a robust tool in drug testing and invasion studies [22,23], and also enable 3D co-cultures with stromal cells [24,25]. However, as with any cell line-based model, spheroids also reflect only a fraction of the molecular diversity found in real tumors.

### 2.2. Patient-Derived Xenografts

In xenograft models, pancreatic cancer cells are planted either ectopically or orthotopically into immunodeficient mice. In patient-derived tumor xenografts (PDXs), tumors are developed from transplanted patient samples. These tumors maintain the tumor architecture, morphology and molecular properties of the primary tumors. More significantly, they can be used in primary PDAC cell line derivation.

PDXs allow for the evaluation of neoplasia progression, ultimately aiming to identify new potential biomarkers. In addition, xenografts can be used as a model for in vivo pharmacodynamic testing, as noted in many studies where drug responses to PDX match patient clinical outcomes [26]. However, PDX models remain expensive and time-consuming, limiting their use in clinical settings [27]. Some genetic drift that is typically not detected in cancer patients has also been observed to occur in PDX models for some tumor types [28].

One alternative to the mouse model is establishing PDX in zebrafish embryos [29]. Zebrafish embryos have a short generation of only several days, carrying a relatively low cost in comparison with PDX in mice. In addition, zebrafish embryos have been used in DSRT for several types of cancers, including PDAC [30,31,32].

### 2.3. Genetically Engineered Mouse Models

Using genetic engineering techniques, specific gene mutations can be induced in primary oncogenes or tumor suppressor genes in mice, leading to the development of pancreatic tumors. This type of model, known as a genetically engineered mouse model (GEMM), enjoys wide use in the study of tumor progression, from precancerous lesions, pancreatic intraepithelial neoplasia (PanINs), to full-blown carcinoma [33,34]. One of the most widely used PDAC GEMMs is the KPC mouse, which generates through pancreas-specific Pdx1-promoter-driven expression of oncogenic KrasG12D and Trp53R172H alleles spontaneous PDAC-like tumors [33,34]. The KPC tumors are hallmarked by heavy fibrosis, and they also metastasize. However, GEMM is expensive and time-consuming. Additionally, tumors are typically of a murine origin, creating caveats when translating findings to human cancer. GEMM tumors grow from cells carrying a limited number of defined mutations, which can be desirable from a mechanistic perspective. It remains unclear how well GEMMs mirror the gradual oncogenic processes and intermittent tumor pruning present in cancer patients. Concerns have been raised that immunoediting, in particular, may be lacking in the KPC model [35].

## 3. Pancreatic Cancer Organoids

### 3.1. Organoid Technology

Cell lines have proved valuable as platforms for disentangling molecular mechanisms, although the quest for physiological models truly predictive of clinical responses in patients continues. Organoid technology, first introduced in 2009, relied on mouse small intestinal cells cultured from stem cells [36]. This was followed by the development of PDOs derived from pancreatic tissue in 2013 [37], and subsequently from other murine and human gastrointestinal epithelial tissues [38]. In 2015, human pancreatic cancer PDOs were first established as a result of collaboration between the Clevers and Tuveson laboratories [18]. The PDO process begins with the creation of a single- or few-cell suspension of tumor tissue using enzymatic digestion and/or mechanical dissociation. Cells are embedded in a physiological matrix, where they expand into multicellular structures. In the case of healthy cells, organoids often become hollow with a clear luminal–apical arrangement, potentially resembling the ductal structures of the pancreas, while cancer cells form both compact and hollow structures of varying sizes [37,39].

In order to achieve mesenchymal signaling, PDOs are cultured in serum-free media with supplemental growth factors that activate the Wnt–Lgr5–Rspo axis, which leads to the proliferation and unlimited expansion of PDAC cells [37]. Supplemental growth factors in culture medium consist of EGF, FGF10, Rspo1, Noggin, Wnt3a, nicotinamide, N-acetylcysteine, gastrin, B27 supplement and A83-01. Additionally, normal human pancreatic PDOs require prostaglandin E2 [40]. In protocols developed by the Muthuswamy and Skala groups, PDOs were cultured in media lacking Wnt ligands [41,42]. Interestingly, Sato et al. [43] discovered both Wnt niche-dependent and -independent PDO subpopulations, along with a Wnt independency correlating with tumor progression. This suggests that using a Wnt ligand-free culture medium preselects more aggressive PDAC organoid subpopulations.

The protocol developed by the Clevers and Tuveson groups enabled the long-term propagation of PDOs [44]. The primary limitation of this protocol is the disappearance of stromal cells after some passages, resulting in ductal cell monocultures. This makes it harder to model interactions between stromal and ductal cells, and tumor microenvironment studies require adding the stromal and immune cells separately to co-cultures. Furthermore, the clonal heterogeneity in cancer organoids may change with passages and should be checked against the tumor sample to ensure representativeness. In comparison, the protocol by the Skala group enables the preservation of fibroblasts in PDO cultures [42]. Moreover, Kuo reported using an air–liquid interface in the PDO formation from murine cells [45].

### 3.2. Organoid Culture Conditions and Success Rates

Several characteristics of PDO cultures continue to complicate organoid isolation and successful propagation. First, a Wnt-conditioned medium, needed for the preparation of an organoid culture medium, is traditionally prepared by Wnt-overexpressing cells. This preparation method results in a high variation between batches, while the activity of Wnt should be measured as a quality control. Additionally, the secretion of active Wnt requires fetal bovine serum, which is detrimental to organoid cells. Recently, the glycoprotein Afamin was found to form a stable complex with Wnt proteins [46]. This discovery led to the production of a serum-free Wnt-conditioned medium featuring more stable concentrations of active Wnt3a. Another study reported the stabilization of Wnt3a using soluble lipid carriers [47].

Beyond improving the culture medium, another means of improving PDO cultures lies in the matrix used. Generally, basement membrane-based matrices (Matrigel or Cultrex) are used. However, the animal origin and variation between batches result in problems in their usage. That said, new gel matrices are being developed, including hydrogels derived from pancreatic tissue [48]. Some studies have reported the successful use of hydrogels in PDO cultures [49,50].

Despite these problems concerning PDO culture conditions, success rates related to deriving long-term cultures over several passages remain better than those for 2D cell lines. In an initial study reporting the establishment of PDOs, the Clevers and Tuveson laboratories achieved success rates of 75% and 83%, respectively, for organoid isolation [18]. Two failures to establish organoid cultures were ascribed to a high stromal cell proportion and fibrosis of the tumor sample in one case and extensive necrosis following neoadjuvant therapy in another case. In a subsequent study conducted by the Tuveson group, a success rate of 78% was reported for surgically resected specimens [51]. In a recent study by Georgakopoulos et al. [50], PDOs were cultured in an optimized serum-free medium resulting in a success rate exceeding 90%.

PDOs have also been established from fine-needle biopsies, whereby one study reported an initial success rate of 87% in establishing primary organoid cultures, followed by attrition from unknown causes resulting in an ultimate long-term success rate of 66% for organoids beyond five passages [52]. This represented a rather significant achievement since, previously, organoids were primarily established from surgical specimens, which contain a much greater number of cells. In a study by Seppälä et al. [53], a 77% success rate was achieved for surgically resected samples and 78% of endoscopic ultrasound-directed core-needle specimens succeeded.

## 4. Organoid Applications

### 4.1. Drug Sensitivity and Resistance Testing and Personalized Medicine

One interesting application of PDOs lies in their use in screening for tumor-specific drug sensitivities and resistance phenotypes, also known as pharmacotyping. The resulting drug profiles are within the sensitivity limits of the assay, often unique for each sample, which generates drug response “fingerprints”. DSRT of PDOs established from surgical resection specimens could theoretically aid clinicians in deciding between different chemotherapy regimens for adjuvant therapy. Furthermore, PDOs isolated from biopsies facilitate the discovery of novel treatments for metastatic cases and for neoadjuvant patients.

Driehuis et al. [54] performed high-throughput DSRT for a panel of 76 therapeutics on 24 PDAC organoids. Differences in responses were detected between different therapeutics and between different patients. Only four patients had sufficient clinical data to identify similarities in therapeutic responses in PDOs and clinical settings. Encouragingly, in these four cases, the clinical data matched the drug responses of PDOs. Tiriac et al. performed a retrospective comparison of organoid responses to five chemotherapeutic agents to progression-free survival (PFS) in nine evaluable patients. Of these patients, six had a greater PFS than historical controls, and organoids from these patients responded well to at least one of the tested drugs, while two PDOs resistant to all tested drugs mirrored clinical treatment resistance [51]. The Muthuswamy and Hidalgo groups described a 96-well-based PDO DSRT platform in which organoids established from both primary tumor tissue and metastases from 12 PDAC patients were tested for responses to several anticancer agents and compared to clinical responses [55]. The authors documented promising concordance between the most or least active single agents classified in two tiers by dose–response area-under-the-curve (AUC). Seppälä et al. [53] performed pharmacotyping for PDOs derived from core-needle biopsies and resected tumors. The mean time to pharmacotyping was 49 days, a sufficient timescale for the use of drug screening results in clinical decision making. Hou et al. [56] reported a semi-automated bioprinting-based method, allowing the rapid seeding of cells to create uniform organoid cultures in 384-well plates for high-throughput screening. These studies provide hope that PDO drug screening can enjoy use in neoadjuvant or metastatic settings.

A recent study Raghavan et al. highlighted the need to better understand the associations between transcriptional identities and drug sensitivities/resistance both in PDOs and cell lines, because the identities changed depending on culture conditions [57]. The authors found that switching culture media altered the transcriptional identities of primary patient tumor-derived organoids as well as established PDAC cell lines with a profound impact on sensitivity at least to SN-38 and paclitaxel. Organoids grown in standard cell culture medium without niche factors lost basal-like transcriptional identity [57]. It has been observed that at least the chemotherapy cocktail FOLFRINOX may promote a basal-like transcriptional state in tumor cells in PDAC patients [58]. A study by Osuna de la Peña showed that the transcriptomic state and responses to gemcitabine + paclitaxel or triptolide in pancreatic tumor-derived 3D cultures growing in self-assembling peptide hydrogels supplemented with core ECM proteins were different compared to organoids growing in matrigel [59].

Another interesting application under development is the establishment of PDOs from circulating tumor cells (CTCs), which could allow tumor cell isolation from the blood of cancer patients and the subsequent establishment of PDOs and drug screening without the necessity of invasive sampling. For instance, CTC-derived organoids have been established from lung cancer [60] and prostate cancer [61] cells. In PDAC, spheroid-like clusters of CTCs and immune cells were established and cultured for up to seven days [62].

Despite numerous studies on PDO drug screening, it remains unclear how accurately drug screening results match patients’ clinical responses and if results can translate into clinical decision making. In the study by Tiriac et al. [51], drug screening results from 89% of patients (8/9 PDOs) were consistent with the clinical response. Similarly, Sharick et al. [63] found that drug screening results matched clinical data for all of the patients (7/7 PDOs) in their study. Grossman et al. found concordance between the ex vivo responses of at least one anticancer agent and the clinical treatment regimen in 12/12 cases [55]. In a study comparing organoids from patients who were administered FOLFIRINOX in the neoadjuvant setting to organoids derived from treatment-naïve patients, the ex vivo PDO responses to FOLFIRINOX were weak/absent in 5 of 5 cases of the neoadjuvant-treated group and robust in 4/5 of the treatment-naïve cases [64]. However, these results rely on small cohorts and more extensive clinical studies among larger groups of patients are needed. Furthermore, patients with good responses all received various combination therapies, most already demonstrated as effective in treating PDAC. In addition, patients receiving different treatments were compared with each other within these studies, rendering the interpretation of results difficult. A systematic review of drug screening results for different types of cancers concluded that the clinical validity of PDO drug screening results remains inconsistent across all tumor types and treatments [65]. This highlights the importance of well-planned co-clinical trials with large patient cohorts comparing one treatment type at a time. It may also be necessary to include different PDO growth conditions to preserve the transcriptional subtypes of the primary tumors, which are known to influence drug responses [57,59].

PDOs make it possible to identify new treatments that are not based on small molecule targeted agents. Oncolytic viruses can also infect PDAC organoids in culture and organoid-derived tumors growing as xenografts in immunocompromised mice [66].

### 4.2. Neoplasia Modeling

Organoids have been used in neoplasia modeling and molecular subtyping. Transplanting pancreatic organoids into immunodeficient mice resulted in preneoplastic lesions, PanINs. These lesions can develop into invasive adenocarcinoma [18], and can then be used to identify new potential biomarkers for early events in pancreatic cancer progression. In a recent study by Miyabayashi et al. [67], PDOs were injected into murine pancreatic ducts, where they gave rise to adenocarcinomas of both main transcriptional PDAC subtypes (“basal-like” and “classic”). Unlike pancreatic cancer cell lines, these transcriptional subtypes have also been identified in ex vivo PDO cultures, albeit not reflecting the full transcriptional features observed in patient tumors [43,51,57]. The basal-like subtype is associated with a poor prognosis and is characterized by the expression of *TP63* and other basal markers, whereas the classic subtype is characterized by the expression of GATA6 and other ductal differentiation markers. Interestingly, Seino et al. [43] found that the classic subtype was associated with Wnt niche dependency in PDO cultures.

### 4.3. Co-Cultures

PDAC features an extensive stromal component, constituting up to 90% of the tumor volume and containing the extracellular matrix (ECM), cancer-associated fibroblasts (CAFs), immune cells and blood vessels. In addition, the total ductal cell-type-enriched proteome derived from pancreatic cancer tissue differs markedly from the corresponding ductal-enriched proteome derived from adjacent non-malignant pancreatic tissue [68]. The tumor stroma may act as both a physical barrier impairing drug delivery [69] and play a key role in the development of biochemical drug resistance [57,59,70,71]. Specific CAF subsets were also found to associate with immunotherapy responses [72,73]. Because PDOs generally lack stromal cells [74], numerous studies have focused on developing co-culture models, which could better depict the tumor microenvironment and interactions between ductal and stromal cells. Co-culture models could also aid in drug screening for stroma-targeting agents and immunotherapies.

The development of co-culture models of PDOs and stromal cells is not problem-free. For instance, many components of the PDAC organoid culture medium are detrimental to CAFs and immune cells. Notably, Öhlund et al. successfully cultured murine PDAC organoids with CAFs in a medium containing DMEM and 5% FBS [75]. This may be explained by the discovery that CAFs are responsible for producing Wnt in the absence of a Wnt-containing medium in PDO and CAF co-cultures [43,75]. In the same study, different functional and molecularly distinct subpopulations of CAFs, formerly thought to be a relatively uniform cell type, were discovered. Myofibroblastic CAFs expressed high levels of alpha smooth muscle actin and resided closer to tumor cells, whereas inflammatory CAFs of distant stroma secreted interleukins. In a subsequent study by Biffi et al. [76], IL-1 and TGF-beta signaling were found to play a crucial role in fibroblast differentiation. Later, the discovery of a third CAF subpopulation was reported by Elyada et al. [77]. These cells, termed antigen-presenting CAFs, expressed the MHCII complex and could present antigens to CD4+ T cells. Recently, CD105 has been found to identify two subpopulations of pancreatic fibroblasts suggested to be tumor-suppressive CD105^neg^ and tumor-permissive CD105^pos^, irrespective of myo/inflammatory/antigen presenting status [78].

In addition to co-cultures with CAFs, co-cultures involving immune cells are under development. Organoids enable the development of functional ex vivo immunotherapy assays. T cells are lost faster than other CD45+ cells in typical embedded epithelial organoid cultures from carcinoid tumors [79], but they can be expanded from the blood or from the primary tumor tissue and subsequently co-cultured with epithelial organoids. Organoid cultures growing at the air–liquid interface retain stromal cells and T cells better than embedded organoids, and the addition of IL-2 can increase T cell persistence up to 60 days [80]. Tsai et al. [81] developed a 3D culture model using PDAC organoids, CAFs and T-cells. This co-culture resulted in increased resistance to gemcitabine when compared to PDOs alone. A recent study by Holokai et al. demonstrated using both murine and human autologous multi-cell-type organoid co-cultures that, in addition to checkpoint receptor/ligands, myeloid-derived suppressor cells can interfere with anti-organoid T cell responses [82]. Co-culturing PDAC organoids with endothelial cells revealed that the endothelial cells could support and maintain CD24+/CD44+ cancer-initiating cells in organoids [83]. Furthermore, the authors found that these cancer-initiating cells are enriched in normal PDAC organoids. This emphasizes the need for continued development of co-culturing systems in cancer organoid research.

Three-dimensional organoid co-cultures serve as an attractive model for studying tumor invasion. For instance, Koikawa et al. [84] found that a tumor organoid co-culture with pancreatic stellate cells, precursors to CAFs, resulted in tumor cell invasion to the surrounding matrix. Osuna de la Peña et al. generated peptide-based hydrogels in which the ECM components and stiffness of the gels could be regulated precisely [59]. The authors showed in organoid-stellate-macrophage co-cultures that both ECM components and matrix stiffness influenced transcriptional and protein signatures of EMT of all cell types. The co-cultures adapted transcriptional and histological signatures that, for certain parts, resembled the tumor tissue closer than monotypic cultures or cultures grown only in Matrigel. In general, using organoids to model cancer invasion and metastasis remains in its infancy.

### 4.4. Organoid Biobanks

Collecting and cryopreserving PDOs enables the formation of PDO biobanks. DNA sequencing ensures that PDOs preserve features of the respective primary tumors. Several organoid biobanks have been established from PDAC tumor tissues [54,85]. These biobanks can be used in personalized medicine and may aid in identifying novel driver mutations and linking specific mutations and epigenetic changes to altered drug responses.

## 5. A Single-Institute Experience: Organoids as a Drug Sensitivity Testing Platform for Pancreatic Cancer

### 5.1. Organoid Culture, Drug Administration and Response Quantification

All procedures described herein were approved by the ethical committee (HUS/216/2017 and HUS/230/2019) and adhere to the Declaration of Helsinki. The success rate in our laboratory for establishing early organoid growth is nearly 100%, and roughly half of these can be cultured beyond the third passage for biobanking. The fraction of organoids that can be re-established after biobanking is being evaluated, currently estimated to be between 20% and 50%.

The Institute for Molecular Medicine Finland (FIMM) has made significant advances in high-throughput precision medicine for many cancer types, showcasing the anecdotal translation of ex vivo DSRT results to clinical use [86,87]. Capitalizing on the FIMM DSRT platform, we adapted a PDO culture to 384-well plates, which are amenable to acoustic compound dispensing, luminescence plate readers and automated microscopy. Surgical material from both the tumor region and an adjacent healthy region from 11 patients was processed as described [44]. In some cases, we submitted DNA from dissociated cells of the primary tumor tissue for targeted next-generation sequencing, covering at least the KRAS, TP53, CDKN2A and SMAD4 genes. The patient cohort and the genetic features of primary surgical tissue and resulting organoids, as well as assays performed on the organoids, are summarized in Table 1.

For drug sensitivity and resistance testing, established organoids growing in several Matrigel domes in 24-well polystyrene plates were dissociated and transformed into a single-cell suspension in an organoid-feeding medium. The cell suspension was diluted and mixed at 1:1 with a cold Matrigel stock (8–10 mg/mL) to a final concentration of 5000–10,000 cells per 10 μL. Leaving one or two rows/columns empty to minimize plate effects, each well was seeded with 10 μL of the cold Matrigel/organoid cell suspension and allowed to set at 37 °C for 20 min. The organoid wells were then topped off with 25 μL of feeding medium, and the empty edge wells were filled with phosphate-buffered saline (PBS). The plates were observed under microscope for sufficient organoid growth, typically for 3 to 7 days. On the day of drug dispensing, referred to as day 0, 10 μL of feeding medium was carefully aspirated from the top of each well and replaced with a fresh feeding medium containing 1:500 Celltox Green (Promega, Madison, WI, USA).

In order to support future mechanistic and combination studies, we focused initially on four anticancer agents of clinical and mechanistic relevance: the targeted MEK inhibitor trametinib, which is useful for probing the effects of inhibiting MAPK signaling downstream of mutated KRAS, pancreatic cancer mainstay chemotherapeutics paclitaxel, gemcitabine and the four-drug cocktail FOLFIRINOX (5-FU, oxaliplatin, leucovorin, irinotecan, in fixed molar ratios of 100:7:27:15, respectively). The chemical agents were dispensed acoustically over a 5–7-point logarithmic concentration range using Echo technology (Labcyte, Sunnyvale, CA, USA) into at least two replicate wells per drug per concentration. To test the feasibility of discovering synergistic drug combinations, some organoids were also assayed with two different drugs in a 7 × 7 well matrix and synergy score was obtained by using highest single agent (HSA) reference model, as described [88]. After 5 days, organoids were imaged at 5× phase contrast (organoid contours) and green fluorescence (cell death) using Cytation 5 (Agilent Biotek, Santa Clara, CA, USA). Next, 35 μL of Celltiter Glo (Promega, Madison, WI, USA) was dispensed into each well and luminescence (ATP levels, viability) was measured after 20 min on Cytation 5.

In a few cases, we also tested whether we could increase the resolution of our assay to pick up possible surviving single cells after treatment. For this, on the day of drug dispensing, we added into the culture medium the nuclear labeling agent Nuclight Rapid Red (Essen/Sartorius, Royston Hertfordshire, United Kingdom) at 1:500 final concentration, as well as Celltox Green at 1:500 (Promega, Madison, WI, USA). Organoids were then imaged before addition of the anticancer agents, as well as on day 3 and day 5 post drug addition, by confocal microscopy (Opera Phenix, Waltham, MA, USA) at the FIMM High-Content Microscopy Unit.

Viability data were used in GraphPad Prism to generate a four-parameter logistic sigmoidal curve fit, for which the area under the curve (AUC) was calculated and converted by inversion and scaling into a drug sensitivity score (DSS), ranging from 0 to 50. Cell death data were obtained by measuring the mean green fluorescence in the organoids based on image analysis on Cytation 5 using the following settings: threshold 8000, object size 20–2000 mm and fill holes. These data were also 4PL-curve-fitted in GraphPad Prism, and a DSS score was generated for each condition.

### 5.2. Organoid Viability Responses Show Both Individual and Shared Patterns

We recorded the viability responses of all organoids as individual dose–response curves (Appendix A). To better understand individual and drug-associated response patterns, we plotted the viability as DSS scores in heatmaps (Figure 2A). Of the tested agents, trametinib modestly reduced organoid viability across most cases, while the chemotherapeutics displayed more varied patterns. In terms of viability, some samples such as PO32p3 were selectively sensitive to only one agent, whereas other samples, such as PO80, seemed sensitive to all tested regimens (Figure 2A). Interestingly, organoids from the same patient profiled at different passages (PO27p1 vs. PO27p5, PO37p11 vs. PO37p16) displayed some variation in viability responses (Figure 2A). Comparison of the individual dose–response curves (Appendix A) suggested that the differences are likely not meaningful, but we do not exclude that they could be relevant, arising for example from ongoing subclonal patterning and genetic drift, as has been observed in PDAC PDOs [53,54]. Other studies comparing organoid drug responses at different passages revealed only small differences [51,64].

### 5.3. Measuring Organoid Cell Death Together with Viability Important for Future Clinical Correlation Studies

Most clinical trials in pancreatic cancer have been abandoned due to futility. Numerous mechanistic studies have shown that tumor cells quickly activate feedback signaling and bypass pathways in response to targeted inhibitors that prevent cell death and allow the tumors to continue growing after a short time [12,13,89,90,91,92,93,94,95,96]. In a subset of samples, we therefore incorporated a way of tracking actual cell death by following the accumulation of the cell-impermeable DNA-binding dye Celltox Green. Similar patterns were found in the summarized cell death data (Figure 2B), but when comparing the viability loss and cell death data, it became evident that not all agents were cytotoxic, despite reducing sample viability. Because of the discrepancy between the two assay readouts, we overlayed the cell viability and cell death dose–response curves (Appendix A). For some samples, we saw the expected inverse relationship, meaning that when cells lose viability, measured as a drop in ATP levels using Celltiter Glo, they also die (Figure 2C). However, in other cases, only cell viability was affected without concomitant increase in cell death. For example, PO77 exposed to gemcitabine displayed a clear drop in cell viability, which was not accompanied by an increase in cell death (Figure 2C). This discrepancy was also evident when inspecting the organoids under microscope (Figure 2D). Organoids are being explored as potential predictive functional assay platforms to guide precision medicine for pancreatic cancer patients [51,53,54,55,64]. The number of studies assessing clinical responses in both patients and organoids is rising but is still too small to draw conclusions about the robustness and reliability of organoids as predictive tests. However, in light of our findings, when such studies are conducted, it would be important to include an actual cell death readout, as organoid growth inhibition (without cell death) can potentially be a source of false positive predictions.

### 5.4. The 384-Well Organoid Profiling Platform Is Useful for Drug Combination Studies

To assess the potential of combining the agents to overcome the resistance to one of the agents and possibly to generate synthetic lethality, we performed a limited two-compound combination matrix feasibility experiment. In our example, the MEK inhibitor trametinib generated strong synergy when combined with paclitaxel in a 7 × 7 well matrix in PO34T organoids, with synergy calculations performed using the FIMM SynergyFinder tool [88] (Figure 2E). Trametinib and other MAPK inhibitors have been tested in pancreatic cancer patients in combination with the conventional chemotherapeutics gemcitabine and paclitaxel but have so far not displayed breakthrough efficacy in large patient cohorts [90,95,96]. However, isolated responses have been observed, which supports the exploration of individual treatment combinations. It may be possible to discover genetic or transcriptional hallmarks of increased sensitivity [97]. Additionally, signaling pathway inhibition by targeted agents can be useful in probing cancer resistance mechanisms and pushing tumor cell signaling toward a state of increased sensitivity to other inhibitors before bypass pathways are activated [89,98,99]. Our tests clearly show that the organoid platform is robust enough to allow the detection of synergistic drug concentrations.

### 5.5. Single Cell Organoid Drug Sensitivity and Resistance Testing Reveals Single Surviving Tumor Cells

Our data show that organoids display individual responses to several anticancer agents, which reflects their different cell states, signaling and resistance mechanisms. We also know that tumor responses in clinical settings are transient at best. Recently, organoids were used to model adaptive resistance to FOLFIRINOX, which was associated with differential transcriptional reprogramming [100]. To establish if organoids can allow us to study how tumors escape even harsh cytotoxic chemotherapies and other treatments, we tracked individual cells in organoids over several days by confocal microscopy using a live cell nuclear labeling agent. Our data showed remarkably accurate quantitation of single cells both in intact organoids and in debris after some of the most potent drug combinations (Figure 3, Appendix A). This method should allow tracking of the true depth of treatment responses. We also tested the feasibility of extracting viable single cells from the gels after treatment for downstream experiments and analyses. So far, we have been able to regrow individual organoids from surviving cells after exposure to trametinib or paclitaxel but not to the combination of these agents for PO34T, PO83T and PO84T, exemplified for PO83T in Appendix A. Using a very high starting number of organoids, estimated to contain more than one million individual epithelial cells, we managed to obtain the required minimum of 1000 live cells for single cell RNA sequencing also from the combination treatments in these three cases. Analysis of the transcriptomic profiles of these cells is ongoing. Our findings support continued studies to understand the nature of tumor cell escape, e.g., by single-cell RNA/DNA sequencing.

## 6. Conclusions and Future Directions

Here, we summarized several interesting and highly relevant characteristics of organoids as model systems for pancreatic cancer research and for the development of new pancreatic cancer treatments. Several benefits of organoids compared with previous models lie in their relative robustness, affordability and faithfulness to mimicking actual disease. Organoids do not yet fully mirror the transcriptional profiles of tumor cells in patient tumor tissue, but work is ongoing to improve the culture conditions and to detail if and how the differences might impact treatment responses [57,59]. Several exciting variations to PDO cultures are under development and the tumor stroma are being incorporated in an increasing number of studies. At the same time, organoids offer a way to enrich tumorigenic cells that can be transplanted into animals, including zebrafish, which may become increasingly useful to validate, e.g., ex vivo PDO drug screening findings.

Pancreatic cancer can only be cured in a small fraction of cases through a combination of surgery and aggressive chemotherapy, which itself is dramatic and carries high intrinsic risks. Organoid cultures are being evaluated by us and others as guidelines for the choice of chemotherapy regimens and for entirely new drugs and combinations. Our own initial findings using monotypic organoid cultures indicate that we may benefit from, including multiple readouts in PDO drug sensitivity and resistance testing. Other groups use a variety of readouts, but the number of organoids and matched patient responses are too low to draw conclusions about the impact of assay parameters on predicting/guiding personalized treatments [51,53,54,55,64]. Clearly, we must continue to test and validate organoids as a predictive platform for clinical efficacy. Even if correlating a clinical benefit through simple assay readouts and monotypic cultures from only single anatomical sites remains difficult, this is essential work on the path toward effective precision medicine for pancreatic cancer. The next wave of studies anticipated in the near future will undoubtedly shed further light on which model systems and assays carry the greatest predictive power.

Several published studies have described the composition and transcriptional evolution of PDAC organoids and primary tumors at a single-cell resolution [57,71,74,101]. However, these studies have not yet revealed the extent to which subclonal resistance mechanisms contributing to tumor escape occur and could be identified at baseline or emerge as a response to treatment. Our own findings indicate that even after harsh drug combinations, individual tumor cells may survive and even regrow once the drugs are withdrawn. Cancer cells adapt and become resistant to sublethal doses of anticancer agents, as was shown for FOLFIRINOX in PDOs [100], but would it be possible to predict which cells or subclones are going to survive even before treatments are started, e.g., using lineage-tracing? Further, would combining transcriptome, epigenome, proteomic and genome data, possibly at the single-cell and spatial level with advanced organoid models including necessary stromal components, really reveal the relevant mechanisms and evolution of drug resistance, or will the wealth of data rather confound efforts to develop effective solutions for pancreatic cancer patients? Like many other groups, we rely on obtaining a mechanistic understanding and functional tools to guide treatment decisions simultaneously. We also aim to use computational models to extract the relevant clues to tackle pancreatic cancer drug resistance. It will be quite interesting to see how intrinsic and adaptive tumor resistance to treatments can be prevented, circumvented, and handled in the future.

## Figures and Tables

**Figure 1 cancers-14-00525-f001:**
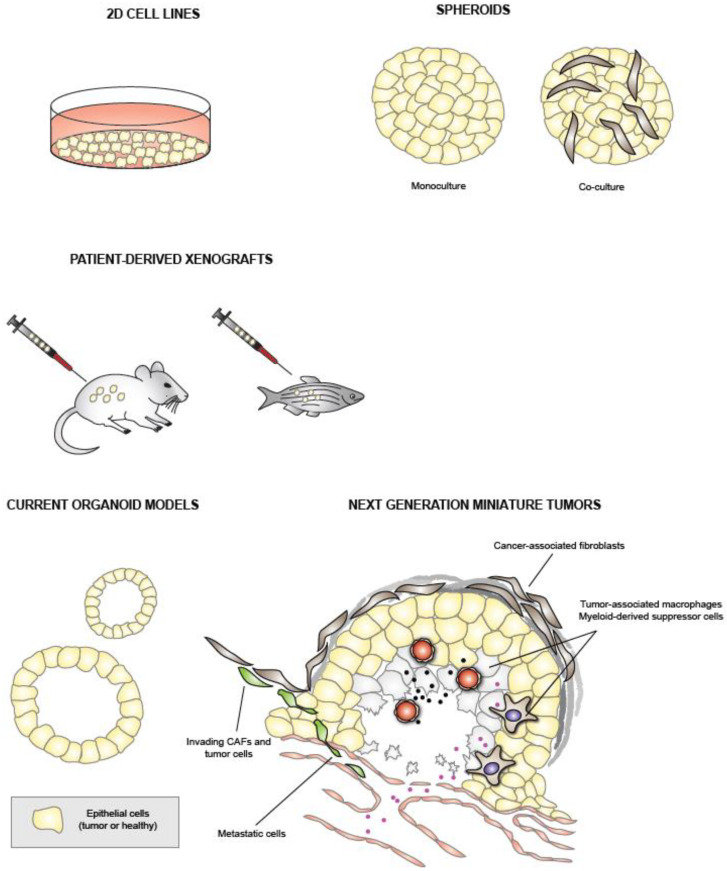
Summary of past and current experimental pancreatic cancer models for development of new treatments. Cell lines constitute the simplest models, while PDOs have begun to replace them in drug sensitivity and resistance testing studies. Several groups are designing controlled multi-cell-type organoid co-cultures, miniature tumors, with the aim of mimicking/maintaining the molecular tumor features as accurately as possible.

**Figure 2 cancers-14-00525-f002:**
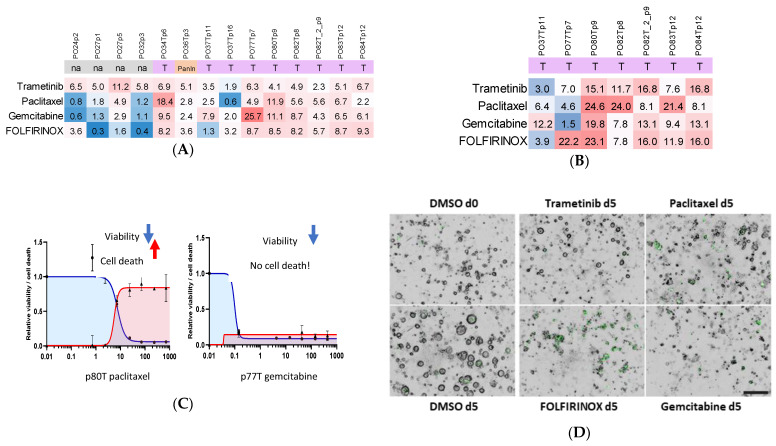
A new organoid drug sensitivity and resistance testing platform. Fourteen organoids from 11 patients were profiled for viability and cell death responses to four anticancer agents (MEK inhibitor trametinib, chemotherapeutics gemcitabine and paclitaxel, and chemotherapy cocktail FOLFIRINOX). (**A**) Summary of cell viability responses, shown as a drug sensitivity score, which was calculated as an adjusted inverse of the area under the curve shown for each sample and drug in Appendix A. Scale 0–50. The table shading reflects the DSS values, with an arbitrary color shift from blue (poor response) to red (good response) around DSS 3 (**B**) Some organoids were also monitored for cell death by fluorescence microscopy. The cell death drug sensitivity score reflects the area under the curve of the dose–response curves shown in Appendix A and an example of quantification in Appendix A. Scale 0–50, color threshold DSS 7. (**C**) Overlay of the cell viability and cell death dose–response curves showed that, in some organoids, select anticancer agents caused both viability loss and cell death (example PO80T), whereas in other organoids only viability was reduced but no cell death was induced (PO77T). (**D**) Low magnification (5×) fluorescence microscopy was used to distinguish cell growth arrest from cell death (green signal) in PO77T organoids. In the example images, FOLFIRINOX is killing the organoid cells, whereas trametinib and gemcitabine have only halted organoid growth as compared to DMSO, in which organoids grow over the 5-day assay. Scale bar = 100 mm. (**E**) The organoid platform is amenable to two-drug combination testing. Two drugs are combined in a 7 × 7 dose-response matrix and viability and cell death are measured. The example shows synergy for trametinib and paclitaxel in reducing sample viability (PO83T), seen as red shading.

**Figure 3 cancers-14-00525-f003:**
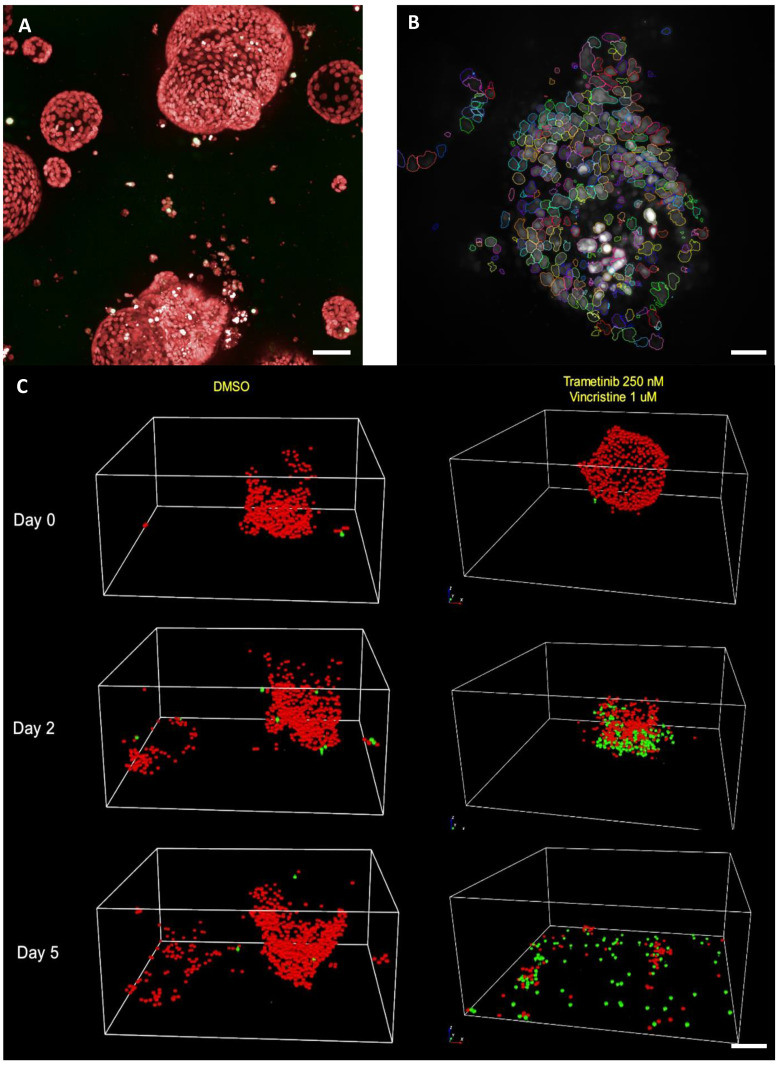
Confocal microscopy allows single-cell drug response monitoring. (**A**) Nuclight Rapid Red reagent allows labeling of individual nuclei in organoids growing in Matrigel; 20× magnification, scale bar = 40 mm. (**B**) Opera Phenix Harmony 3D segmentation pipeline distinguished the individual nuclei, seen as colored outlines. Scale bar = 20 mm. (**C**) PO83T organoids were tracked for 5 days. DMSO (control) organoids increase in size, while the combination treated organoids were destroyed. Importantly, the live nuclei (red) and cell death (green) labeling reagents allow monitoring of single surviving cells. Scale bar = 30 mm. Video in Appendix A.

**Table 1 cancers-14-00525-t001:** In most cases, the tissue that was processed for organoid culture was obtained from the tumor region, and the diagnosis was pancreatic ductal adenocarcinoma. Variant allele frequencies for KRAS, TP53 and other oncogenes were obtained using next-generation sequencing of DNA extracted from the primary surgical tissue and/or the established organoids (FIMM Technology Center and HUSlab, under process). Organoid passage numbers at the time of the drug sensitivity assay are shown. Some organoids were profiled only for viability responses using Celltiter Glo (CTG), while others were also assessed for cell death (Celltox Green, CTX).

Case					Mutations						Drug Screen			
ID	Surgical Site	Diagnosis	TNM	Grade	KRAS			TP53		Others	Final ID	Passage	Readout	
					Tissue	Organoid	Codon	Tissue	Organoid				CTG	CTX
PO24	Tumor	PDAC	pT3N1	G2	na	na	na	na	na	na	PO24	p2	1	0
PO27	Tumor	PDAC	pT2N1	G2	na	na	na	na	na	na	PO27	p1	1	0
												p5	1	0
PO32	Tumor	PDAC	pT3N1	G2-3	na	na	na	na	na	na	PO32	p3	1	0
PO34	Tumor	PDAC	pT2N0	G2	46.8	47	G12D	100	100	CDKN2A-/-, SMAD4-/-	PO34T	p6	1	0
PO36	Tumor	PanIN2	na	na	na	48	G13D	0	0	0	PO36PanIN	p3	1	0
PO37	Tumor	PDAC	pT3N2	G2	0	100	G12D	0	100	CDKN2A-/-	PO37T	p11	1	1
												p16	1	0
PO77	Tumor	PDAC	pT3N1	G3	48.7	64.1	G12D	38.6	97.7	0	PO77T	p7	1	1
PO80	Tumor	PDAC	pT2N1	G2	na	66.3	G12D	na	100	0	PO80T	p9	1	1
PO82	Tumor	PDAC	pT2N1	G2	na	58	G12R	na	99	CDKN2A-/-	PO82T	p8	1	1
	Adjacent			G2	na	72	G12R	na	98	0	PO82T_2	p9	1	1
PO83	Tumor	PDAC	pT3N2R1	G2	na	69	G12D	na	100	0	PO83T	p12	1	1
PO84	Tumor	PDAC	pT2N1	G2	na	48	G12D	na	na	0	PO84T	p12	1	1

## Data Availability

Not applicable.

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
