# Peer review of "Pancreatic Cancer Organoids in the Field of Precision Medicine: A Review of Literature and Experience on Drug Sensitivity Testing with Multiple Readouts and Synergy Scoring"

_cancers, 2022, doi:10.3390/cancers14030525_

Round 1

Reviewer 1 Report

This is short mini review about Precision medicine of PDAC. This topic is interesting. This paper reviews the current PDAC model systems, and focus on PDO models and their applications. They then present in Section 5 the technical development of and initial results from our own pancreatic cancer organoid based DSRT. However, this manuscript should also review several important word in section 5 from other group, instead of only themselves. Besides, there is no figure caption for each figure. Furthermore, the author should give full discussion of Precision medicine of PDAC in “Future Directions”

Author Response

Reviewer 1

This is short mini review about Precision medicine of PDAC. This topic is interesting. This paper reviews the current PDAC model systems, and focus on PDO models and their applications. They then present in Section 5 the technical development of and initial results from our own pancreatic cancer organoid based DSRT.

  1. However, this manuscript should also review several important word in section 5 from other group, instead of only themselves.
  2. Besides, there is no figure caption for each figure.
  3. Furthermore, the author should give full discussion of Precision medicine of PDAC in “Future Directions”

RESPONSES:

  1. We thank the reviewer for pointing this out and have added some more comparisons to work published by others. We have referenced in section 4.1 some of the largest organoid-based drug screening studies conducted in pancreatic cancer to date, and as we point out in section 6, the readouts in these studies differ from each other. We also acknowledge the current limitation of these studies in that the number of samples is still relatively small, which limits the statistical power of formal testing of the assay parameters in predicting clinical responses.
  2. Indeed, we apologize for this omission! For some reason the figure legends were missing in the first version, but we have now added them into the main text after each figure.
  3. Thank you for the suggestion. We have expanded on the discussion by including some of the issues we feel are particularly important/burning. In addition, we included more references throughout the main text that highlight the current status of pancreatic precision medicine, including the summary of the Know Your Tumor PDAC precision medicine trial (Pishvaian et al., 2020, Lancet Oncol), line 59.

Reviewer 2 Report

Mäkinen et al presented a review of the applications of different pancreatic cancer models for drug sensitivity testing with a focus on a newly emerged model – pancreatic cancer organoids. The topic presented by this manuscript is timely and useful, however, several points need to be addressed prior to acceptance of publication.

  1. In several places, texts need to be modified to reflect accurate or updated knowledge of the human pancreatic cancer research field. For example, 5-year survival rate of PDA has been updated in 2021. Citation should be updated.
  2. In fact, previous publications have suggested that patient-derived immune cells co-exist in early passages of PDX. Text should be modified accordingly.
  3. Provide accurate full name of PanIN.
  4. In 4.2. Neoplasia Modeling section, the paragraph describing the study from Raghavan et al., (2021, Cell) is not related to neoplasia modeling. This reference fits better with section 4.1.
  5. Fig 2, authors stated that PO27 were selectively sensitive to only one agent, however, PO27 PO27p1 and p5 appear to have different response pattern.
  6. Line 459: Figure 1D should be 2D. It’s unclear what this figure is showing - providing Figure legends would be helpful.
  7. Data shown in Figure 3 could be novel and original, however, they are not sufficiently explained in text. Again, Figure legends should be provided.
  8. The authors stated that “When developing co-culture models for therapeutics, it is essential to match a patient’s PDOs and stromal cells with each other in order to achieve reliable results for drug responses”. However, it’s unclear where this conclusion was drawn from. Evidence of whether patient matched CAFs are superior to tumor educated CAFs from other sources should be provided.

Author Response

Reviewer 2

Mäkinen et al presented a review of the applications of different pancreatic cancer models for drug sensitivity testing with a focus on a newly emerged model – pancreatic cancer organoids. The topic presented by this manuscript is timely and useful, however, several points need to be addressed prior to acceptance of publication.

  1. In several places, texts need to be modified to reflect accurate or updated knowledge of the human pancreatic cancer research field. For example, 5-year survival rate of PDA has been updated in 2021. Citation should be updated.
  2. In fact, previous publications have suggested that patient-derived immune cells co-exist in early passages of PDX. Text should be modified accordingly.
  3. Provide accurate full name of PanIN.
  4. In 4.2. Neoplasia Modeling section, the paragraph describing the study from Raghavan et al., (2021, Cell) is not related to neoplasia modeling. This reference fits better with section 4.1.
  5. Fig 2, authors stated that PO27 were selectively sensitive to only one agent, however, PO27 PO27p1 and p5 appear to have different response pattern.
  6. Line 459: Figure 1D should be 2D. It’s unclear what this figure is showing - providing Figure legends would be helpful.
  7. Data shown in Figure 3 could be novel and original, however, they are not sufficiently explained in text. Again, Figure legends should be provided.
  8. The authors stated that “When developing co-culture models for therapeutics, it is essential to match a patient’s PDOs and stromal cells with each other in order to achieve reliable results for drug responses”. However, it’s unclear where this conclusion was drawn from. Evidence of whether patient matched CAFs are superior to tumor educated CAFs from other sources should be provided.

RESPONSES:

  1. We thank the reviewer for the suggestion. We’ve included (line 39) our own recent register study, which reflects an interesting increase in overall survival during the past decade compared to the one before. Several recent publications have been added throughout the manuscript.
  2. Indeed, we agree and added relevant references and modified the text, section 4.3, line 375-387.
  3. Done, line 151.
  4. Yes, we agree and moved it.
  5. We thank the reviewer for making this observation. We now highlight the results from the organoids from the same patient at different passages, section 5.2, line 534, and include discussion in the same section.
  6. We corrected the mistake and added figure legends.
  7. Yes, we now include figure legends and apologize for missing them in the first version.
  8. We decided to strike this statement, the topic is being studied but appears contested and complex and therefore outside the scope of this review.

Reviewer 3 Report

The section on GEMM is rather short and lacks information on specific models and their respective pros and cons. Can the authors elaborate on that?

The section on drug sensitivity assays and resistance assays describes multiple studies and their respective outcomes. However, no explanation is provided  why these studies have varying results and why they do not always match the clinical reality. Do the authors have a hypothesis why this is the case?

It would greatly enhance the clarity of the manuscript if the authors could provide figure legends next to or underneath the figures shown in the manuscript. 

The authors tracked surviving cells after cytotoxic therapy. Do the authors have any more information on these cells besides the fact that they did survive? Do you have quantitative data? What characteristics do these cells have?

The authors describe that using a MEK inhibitor in combination w/ paclitaxel showed "strong synergy". What does synergy mean in this context? Decreased cell viability? Furthermore, the authors conclude in this section that their platform is robust enought to detect synergistic drug combination while a couple sentences earlier they concede that MEK inhibitors are not efficient in patients. Would a more balanced conclusion perhaps fit the data better?

Author Response

Reviewer 3

  1. The section on GEMM is rather short and lacks information on specific models and their respective pros and cons. Can the authors elaborate on that?
  2. The section on drug sensitivity assays and resistance assays describes multiple studies and their respective outcomes. However, no explanation is provided why these studies have varying results and why they do not always match the clinical reality. Do the authors have a hypothesis why this is the case?
  3. It would greatly enhance the clarity of the manuscript if the authors could provide figure legends next to or underneath the figures shown in the manuscript.
  4. The authors tracked surviving cells after cytotoxic therapy. Do the authors have any more information on these cells besides the fact that they did survive? Do you have quantitative data? What characteristics do these cells have?
  5. The authors describe that using a MEK inhibitor in combination w/ paclitaxel showed "strong synergy". What does synergy mean in this context? Decreased cell viability? Furthermore, the authors conclude in this section that their platform is robust enought to detect synergistic drug combination while a couple sentences earlier they concede that MEK inhibitors are not efficient in patients. Would a more balanced conclusion perhaps fit the data better?

RESPONSES:

  1. We thank the reviewer for the suggestions. The topic in section 2.3 is not the focus of our review, but we included more details and references on the most popular model, KPC.
  2. We included more discussion on the topic. The references in section 4.1 show very promising correlation between clinical and organoid responses. The definition of responses in these studies are somewhat arbitrary, at least sometimes very close to the classification thresholds. Overall, however, the number of matched clinical- and organoid response profiles is still small.
  3. As addressed in the previous responses, we added the missing figure legends. Our sincere apologies for this inconvenience.
  4. We included example data describing regrowth of the drug-treated cells, Supplementary Data 5. This will be very useful for downstream experiments aiming to uncover mechanisms of tumor escape. In our ongoing work we are characterizing these cells by single cell RNA-sequencing after exposure to two drugs and the combination. We have already seen by unsupervised clustering (tSNE, UMAP) of three cases (PO34T, PO83T, PO84T) that there are transcriptional subclones even at the (non-treated) baseline in all three cases. All three cases differ markedly from each other, but we find many shared drug-specific features. We would like to keep this preliminary data for future publications.
  5. We have revised the text and included a descriptive figure legend. We also changed the related discussion in section 5.4, line 586.

Reviewer 4 Report

Mäkinen et al are reporting about the use of organoid cultures to enable advanced drug testing for PDAC. The introduction is detailed and very well written describing the current treatment, time point of diagnosis and need for advanced or more precise therapies in this devastating disease. They are as well describing the different models to achieve this goal. Furthermore, they are describing the usefulness of organoids for drug sensitivity and resistance testing. Lastly, they are reporting about their own experiences using organoids for drug sensitivity testing with implementation of quantified cell death. The data is described in a very profound and detailed way that makes it easy to follow. Thank you for this! 

The study is overall very good to excellent and I only have minor comments and suggestions:

The methods are well described. However, it would help to have some more details like concentrations of drugs used, cell counts etc.

Could cell death celltox green be better quantified using FIJI e.g. as appears very subjective?

Scale bars missing  Fig 2D

Figure 2E right figure (synergy finder) hard to interpret due to size

Figure 3 scale bars missing

I do not really understand why some references (?) numbers in square [77] and others name and year and journal?

Author Response

Reviewer 4

Mäkinen et al are reporting about the use of organoid cultures to enable advanced drug testing for PDAC. The introduction is detailed and very well written describing the current treatment, time point of diagnosis and need for advanced or more precise therapies in this devastating disease. They are as well describing the different models to achieve this goal. Furthermore, they are describing the usefulness of organoids for drug sensitivity and resistance testing. Lastly, they are reporting about their own experiences using organoids for drug sensitivity testing with implementation of quantified cell death. The data is described in a very profound and detailed way that makes it easy to follow. Thank you for this!

The study is overall very good to excellent and I only have minor comments and suggestions:

  1. The methods are well described. However, it would help to have some more details like concentrations of drugs used, cell counts etc.
  2. Could cell death celltox green be better quantified using FIJI e.g. as appears very subjective?
  3. Scale bars missing Fig 2D
  4. Figure 2E right figure (synergy finder) hard to interpret due to size
  5. Figure 3 scale bars missing
  6. I do not really understand why some references (?) numbers in square [77] and others name and year and journal?

RESPONSES:

  1. We thank the reviewer for the suggestions and added figure legends and concentrations where needed. Cell counts are assay format-specific and scalable.
  2. Supplementary Data 2 shows the quantitative cell death data, while we added in Supplementary Data 3 the microscopy image analysis masking from which the quantitative data was derived. Low resolution microscopy was done using 5x objective on Cytation 5, with the method described in section 5.1, line 472.
  3. We added scale bars.
  4. We enlarged the panel. We stand ready to adjust the size of any other panels to better fit overall page layout.
  5. As in 3.
  6. We thank the reviewer for noticing this and have made the necessary correction.

Round 2

Reviewer 1 Report

This manuscript could be accepted after this revision